# How Cardiac Fibrosis Assessed via T1 Mapping Is Associated with Liver Fibrosis in Patients with Non-Alcoholic Fatty Liver Disease

**DOI:** 10.3390/jcm12237381

**Published:** 2023-11-29

**Authors:** Flavia Vernin de Oliveira Terzi, Gabriel Cordeiro Camargo, Daniella Braz Parente, Ana Maria Pittella, Gilberto Silva-Junior, Gabrielle Gonçalves de Novaes, Jaime Araújo Oliveira Neto, Julia Machado Barroso, Martha Valéria Tavares Pinheiro, Adriana Soares Xavier de Brito, Renée Sarmento de Oliveira, Rosana Souza Rodrigues, Renata de Mello Perez, Andréa Silvestre de Sousa, Renata Junqueira Moll-Bernardes

**Affiliations:** 1D’Or Institute for Research and Education—IDOR, Rio de Janeiro 22281-100, Brazil; flavia.terzi@riosdor.com.br (F.V.d.O.T.); gabriel.camargo@idor.org (G.C.C.); daniella.parente@idor.org (D.B.P.); ana.pittella@quintador.com.br (A.M.P.); gilberto.ajunior@quintador.com.br (G.S.-J.); gabrielle.novaes@quintador.com.br (G.G.d.N.); jaime.neto@rededor.com.br (J.A.O.N.); julia.barroso@idor.org (J.M.B.); martha.pinheiro@rededor.com.br (M.V.T.P.); medicina.nuclear@inc.saude.gov.br (A.S.X.d.B.); renee.oliveira@unirio.br (R.S.d.O.); rosana.rodrigues@idor.org (R.S.R.); renata.perez@idor.org (R.d.M.P.); andrea.silvestre@fiocruz.br (A.S.d.S.); 2School of Medicine, Federal University of Rio de Janeiro, Rio de Janeiro 21941-853, Brazil; 3Evandro Chagas National Institute of Infectious Diseases, Oswaldo Cruz Foundation, Rio de Janeiro 21040-360, Brazil

**Keywords:** nonalcoholic fatty liver disease, cardiovascular disease, fibrosis, myocardial extracellular volume, magnetic resonance

## Abstract

(1) Background: Nonalcoholic fatty liver disease (NAFLD) is one of the most common chronic liver diseases worldwide. Although cardiovascular and NAFLD risk factors overlap, an independent association between these conditions may exist. Hepatic and cardiac fibrosis are important markers of mortality, but the correlation between these markers in patients with NAFLD has not been well studied. Our main objective was to determine the degree of myocardial fibrosis in patients with NAFLD and its correlation with the severity of liver fibrosis. (2) Methods: In this cross-sectional study, patients with NAFLD were allocated to two groups according to the stage of liver fibrosis assessed using MRI: no or mild fibrosis (F0–F1) and significant fibrosis (F2–F4). Framingham risk scores were calculated to evaluate cardiovascular risk factors, and patients underwent multiparametric cardiac and abdominal MRIs. (3) Results: The sample comprised 44 patients (28 with no or mild liver fibrosis and 16 with significant liver fibrosis). The mean age was 57.9 ± 12 years, and 41% were men. Most patients had high cardiac risk factors and carotid disease. Relative to patients with no or mild liver fibrosis, those with significant fibrosis had a higher median calcium score (*p* = 0.05) and increased myocardial extracellular volume (ECV; *p* = 0.02). Liver fibrosis correlated with cardiac fibrosis, represented by the ECV (r = 0.49, *p* < 0.001). The myocardial ECV differentiated patients with and without significant liver fibrosis (AUC = 0.78). (4) Conclusion: This study showed that diffuse myocardial fibrosis is associated with liver fibrosis in patients with NAFLD.

## 1. Introduction

Non-alcoholic fatty liver disease (NAFLD) is the world’s leading cause of chronic liver disease, affecting up to 30% of the adult population and having a broad histological spectrum that ranges from steatosis to steatohepatitis and liver fibrosis [1]. NAFLD is frequently associated with visceral obesity, dyslipidemia, insulin resistance, and diabetes, which are components of Metabolic Syndrome [2,3], but current evidence suggests that NAFLD is an independent risk factor for CVD, even after adjustment for other classical comorbidities. Advanced NAFLD may involve the release of pro-inflammatory, vasoactive, and thrombogenic molecules that play important roles in the development and progression of cardiovascular disease (CVD) [2], and researchers have hypothesized that NAFLD is not merely a marker of CVD but is also involved in its pathogenesis [3]. NAFLD is associated with markers of subclinical atherosclerotic disease, such as carotid intimal thickness and coronary calcification [4].

The spectrum of NAFLD involves isolated steatosis and non-alcoholic steatohepatitis (NASH), which has an increased risk of progression to cirrhosis. Liver biopsy is still considered the gold standard to diagnose and grade the severity of steatosis and fibrosis; however, it is invasive and expensive, may require hospitalization, and is not always effective due to sampling variability. The high prevalence and burden of chronic liver disease led to the development of different methods to diagnose and monitor disease progression, including magnetic resonance imaging (MRI) [5]. Advanced MRI techniques, such as magnetic resonance elastography (MRE), can detect increased liver stiffness caused by an increased collagen deposition and extracellular matrix, allowing for an accurate assessment of fibrosis [6].

Similarly, cardiac disease may result in myocardial fibrosis, which is associated with worse prognosis in different cardiac disease [7,8,9]. Diffuse fibrosis in the heart distorts myocardial architecture, culminating in mechanical, coronary vasomotor, and electrical dysfunction. This represents a phenotype of cardiac vulnerability and occurs in a wide variety of conditions—including diabetic and hypertensive heart disease, as well as ischemic and nonischemic cardiomyopathy—as an early marker and important etiological factor for diastolic dysfunction, heart failure with reduced or preserved EF, and sudden death [10]. Cardiovascular MRI, using parametric mapping techniques, is a non-invasive tool for quantifying tissue alterations in a variety of myocardial diseases. MRI with late gadolinium enhancement (LGE) is a widely utilized non-invasive imaging protocol for myocardial tissue characterization and the identification of replacement fibrosis, but it lacks sensitivity to detect the accumulation of collagen within the extracellular myocardial space. For the evaluation of the reversible, early stage of the diffuse myocardial interstitial fibrosis, MRI uses T1 mapping and extracellular volume (ECV) measurements [11,12,13]. Extracellular volume fraction quantification is associated with prognosis in various cardiac diseases, such as several causes of cardiomyopathies (ischemic, inflammatory, hypertensive, and infiltrative).

Liver fibrosis was associated with heart disease in an arm of the Multi-Ethnic Study of Atherosclerosis (MESA) [14], a populational trial including more than 2000 patients without CVD on admission. Patients who developed CVD, heart failure, and atrial fibrillation within 10 years had more liver fibrosis, as was measured via MRI.

Briefly, interstitial fibrosis in the heart and liver worsens prognoses, and CVD is the principal cause of death in patients with NAFLD; however, to our knowledge, the correlation between liver and cardiac fibrosis has not yet been studied. Our purpose in this study was to assess the presence of interstitial myocardial fibrosis in patients with NAFLD using ECV measurement by MRI with T1 mapping, correlate it with the presence of liver fibrosis, and define its association with the presence of cardiovascular risks, cavitary remodeling, diastolic dysfunction, and coronary calcification.

## 2. Materials and Methods

### 2.1. Patients

Adult (age ≥ 18 years) patients with a clinical diagnosis of steatosis or cirrhosis caused by NAFLD, corroborated by ultrasounds, were prospectively included in this cross-sectional study from June 2019 to August 2021. All of the patients were monitored at the hepatology outpatient clinic of Quinta D’Or Hospital, Rio de Janeiro, Brazil. The inclusion criteria were as follows: the presence of steatosis in spectroscopy defined as liver fat fraction > 5.56% or increased liver stiffness via MRE in patients with a previous diagnosis of NAFLD. Patients were excluded if they had neither criterion for steatosis nor increased liver stiffness. Patients were divided into two groups: those with MRE stiffness < 3.5 kPa were classified as having no or mild fibrosis (F0–F1), and those with ≥3.5 kPa were considered to have significant fibrosis (F2–F4). Framingham risk scores [15] were calculated, and all patients underwent liver and cardiac magnetic resonance studies, tomography for coronary calcium score calculation, color Doppler echocardiography, and carotid Doppler studies.

### 2.2. Study Protocol

Framingham risk scores [15] were calculated to estimate the risk of coronary artery disease development within 10 years. This score adequately identifies individuals at high and low risks of myocardial infarction and death from coronary disease. The data needed to calculate it were collected using a clinical–epidemiological form.

A transthoracic two-dimensional echocardiography with color Doppler (Vivid E95, 2.5-Hz probe; GE Med Ultrasound AS, Horten, Norway) was performed according to the recommendations of the American Society of Echocardiography and the European Association of Cardiovascular Imaging [16]. Measurements of diameters and cavitary volumes, ejection fractions (EFs) using the Simpson method [17], and global and segmental contractility of the left ventricle (LV); analysis of mitral flow; and tissue Doppler imaging of the mitral ring were performed. The right ventricle was qualitatively analyzed through tricuspid annular plane systolic excursion and S’ wave measurements of the tricuspid ring. To obtain the global longitudinal strain (GLS) and standard deviation of the time to longitudinal peak strain of 17 cardiac segments, multiple consecutive cardiac cycles of the three standard apical views were acquired and stored digitally, based on guidelines [18,19,20,21]. The best cardiac cycle with good quality and clear endocardial boundaries was chosen, and the endocardial borders were automatically identified and tracked throughout the cardiac cycle. When the images were not optimal, manual adjustments were made. LV GLS measurement and analyses were performed using commercially available software (Image Arena 4.6; Tomtec, Munich, Germany).

Carotid color Doppler imaging (Vivid E95, 4–10 MHz probe; GE Med Ultrasound) was performed according to the recommendations of the American Society of Echocardiography [22]. The analysis included the evaluation of the carotid intima–media thickness (CIMT) and the presence of plaques.

Calcium scores were obtained with a 256-detector row scanner (Revolution CT; GE Healthcare, Milwaukee, WI, USA). Coronary calcium was quantified semi-automatically to calculate the volume, mass, and calcium score using the Agatston method [23]. Values of 0–10 were classified as minimal calcification, those of 11–100 were taken to reflect mild calcification, those of 101–400 were classified as moderate calcification, and values > 400 were taken to indicate severe calcification. We defined cases of moderate and severe calcification as altered.

Multiparametric resonance studies were performed on a 3.0 T MRI system (Magnetom Prisma; Siemens Healthcare, Erlangen, Germany) using a combined eighteen-element phased-array abdominal coil and a thirty-two-element fixed spine coil.

Liver MRI—2D-gradient echo magnetic resonance elastography (MRE)—was performed to assess liver stiffness and magnetic resonance spectroscopy (MRS) for the detection and quantification of liver fat fraction.

MR spectroscopy (HISTO) with stimulated echo acquisition mode (STEAM) provided proton density fat fraction corrected for T2 and fat and water transverse relaxation. Five STEAM spectra were generated at TE 12, 24, 36, 48, and 72 ms; TR, 3000; flip angle 90°; voxel size 30 mm × 30 mm × 30 mm, placed on liver Couinaud segment V, avoiding liver borders, and large vessels and bile ducts. MRS was performed for the evaluation of liver steatosis, defined as liver fat fraction > 5.56%.

MR elastography: a 2D gradient-recalled echo MRE was used to acquire liver elasticity maps. Sequence parameters were as follows: TR, 25 ms; TE, 15.19 ms; flip angle, 12°; FOV, 380 mm; active driver frequency, 60 Hz; voxel size, 1.5 × 1.5 × 6 mm; number of slices, 4 slices; acceleration factor, generalized auto calibrating partially parallel acquisition (GRAPPA) 2. A pneumatic active wave driver and a tube-connected and strap-secured passive driver were placed on the right side of the anterior chest wall at the level of the xiphoid to measure liver stiffness. Generated shear waves at a fixed vibration frequency coursed through the liver and created tissue displacements to generate magnitude and phase images. Slices were centered over the portal vein. The 4 slices were acquired in 4 consecutive breath holds at end expiration.

MRE was performed for the evaluation of steatohepatitis and fibrosis based on liver stiffness [24,25]. Patients were classified into those with no or mild fibrosis (F0–F1) if liver stiffness values were <3.5 kPa, and those with significant fibrosis (F2–F4) if stiffness values ≥ 3.5 kPa [25].

Cardiac MRI images were acquired before and after the intravenous injection of gadolinium-DOTA contrast (Dotarem; Guerbet, Aulnay-sous-Bois, France) according to the following protocol [26]: the acquisition of cine MR images using a steady-state free precession sequence with retrospective gating, the acquisition of delayed enhancement images with an inversion recovery gradient echo sequence [27], and the acquisition of images using the Modified Look-Locker inversion recovery (MOLLI) sequence for myocardial and liver T1 mapping [10,28,29]. A blood sample was collected, and the ECV was calculated using the hematocrit value to adjust for contrast volume distribution.

Cine images were evaluated to determine LV morphological and functional parameters [30]. Qualitative and quantitative characteristics of myocardial replacement fibrosis were visually determined using delayed enhancement images. Diffuse interstitial myocardial fibrosis were inferred from myocardial ECV, which were calculated using the mean left ventricular myocardial T1 value that was measured in pre- and post-contrast T1 maps obtained at a single short-axis plane at the mid-ventricular level [31].

### 2.3. Statistical Analysis

Means with standard deviations or medians with interquartile intervals were calculated for continuous variables. Categorical variables were described as percentages. The kurtosis test was used to examine variable distributions. The Mann–Whitney, chi-squared and Fisher exact tests were used for between-group comparisons as appropriate. Spearman test correlation analysis between cardiac ECVs and liver fibrosis measured by stiffness on the MRE was performed. A receiver operating characteristic (ROC) curve was used to evaluate the ability of ECV to discriminate between patients with and without significant liver fibrosis. For all of the tests, the level of significance was set to *p* < 0.05. The data were analyzed using SPSS (version 21; IBM Corporation, Armonk, NY, USA).

## 3. Results

Of the 54 patients enrolled in the study, 10 were excluded because they did not fulfill the MRI criteria for NAFLD. Thus, data from 44 patients with NAFLD were included in the analysis. The mean age of the patients was 57.9 ± 12.0 years, and 41% were men. A total of 28 (64%) patients had no or mild fibrosis, and 16 (36%) patients had significant fibrosis (Figure 1).

The clinical characteristics of the patients are presented in Table 1. Metabolic abnormalities, including obesity, diabetes, hypertension, and dyslipidemia, and family histories of heart disease, were prevalent in both groups. Most (77.3%) of the patients had high Framingham risk scores, with no significant difference between the groups. The patients had no myocardial, congenital, or valvular heart diseases. Only one patient had coronary disease. The patients had no history of atrial fibrillation, and all of them were in sinus rhythm during echocardiography and MRI evaluations. A total of 11 patients had clinical signs of liver cirrhosis, and all of these were from the group with significant fibrosis.

Regarding previous medications, 21 patients were taking angiotensin II receptor blockers, 3 had angiotensin I-converting enzyme inhibitor, 11 were using β-blockers, 10 patients were taking diuretics, 2 patients were receiving insulin, 32 used oral hypoglycemic drugs, and 21 were taking statins.

The imaging study results are shown in Table 2. The patients had no left or right ventricular global or segmental systolic dysfunction, no significant LV hypertrophy, and no evidence of cardiomyopathy, as assessed using echocardiography and MRI. LV GLS assessed via echocardiography was normal in all patients. Regarding diastolic function, the two groups had similar indexed left atrial volumes, but patients with advanced liver fibrosis had increased mean E/e’ values (10.0 vs. 8.7 for patients without significant fibrosis, *p* = 0.04). Patients had no signs of elevated LV filling pressures or inferior vena cava dilatation. Carotid Doppler evaluation revealed plaques or increased CIMT in most patients, with no significant difference between groups with and without significant fibrosis (81% and 60.7%, respectively, *p* = 0.20). Plaques causing >50% diameter stenosis were identified in only one patient with significant fibrosis. Calcium scores varied widely within and between groups. Patients with significant liver fibrosis had a higher median calcium score (124.5 vs. 11.0 for patients without significant fibrosis, *p* = 0.05). Calcium scores showed a positive correlation with liver stiffness (r = 0.352; *p* = 0.019) but no correlation with steatosis (r = −0.236; *p* = 0.124). There was no association between Framingham risk scores and steatosis (*p* = 0.484).

There was no statistical difference between the LV volumes and mass evaluated via MRI between the groups. Although left atrial volume was not different between the groups, there was increased left atrial volume in 25.0% vs. 43.8% of the patients without and with significant fibrosis (*p* = 0.313). Cardiac MRI with T1 mapping showed that the median ECV was higher for the group with significant liver fibrosis than for the group without such fibrosis (26.0% vs. 22.7%, *p* = 0.002; Figure 2 and Figure 3; Figure 1). Cardiac MRI revealed late-enhancement fibrosis with a meso-epicardial non-ischemic aspect in five patients (three in group without fibrosis and two in the group with significant liver fibrosis). The area under the ROC curve for myocardial ECV differentiating the groups with and without significant liver fibrosis was 0.78 (*p* = 0.002; Figure 4).

The mean stiffness on MR elastography was 2.1 ± 0.4 kPa for patients with no or mild fibrosis and 4.8 ± 1.1 kPa for those with significant fibrosis (Figure 5). The myocardial ECV correlated with the liver fibrosis measured using MR elastography (*r* = 0.49, *p* = < 0.001). Liver fibrosis was also negatively correlated with the GLS (*r* = –0.36, *p* = 0.02) and positively correlated with the left atrial volume (*r* = 0.30, *p* = 0.04).

## 4. Discussion

Most patients with NAFLD enrolled in this study had CVD risk factors and a high risk of developing severe cardiovascular events within 10 years, regardless of the degree of liver fibrosis. Patients with significant liver fibrosis had higher calcium scores and mean E/e’ ratios. Those with significant liver fibrosis had larger myocardial ECVs than those with no or mild liver fibrosis.

Epidemiological information correlates NAFLD with markers of subclinical atherosclerosis, such as increased CIMT and coronary calcium scores, independent of classical risk factors [32]. A cohort study validated the use of the Framingham score in this context, as it can predict the risk of coronary heart disease development at the time of NAFLD diagnosis [33,34]. In our population, patients in both groups had these two powerful markers in addition to high-risk Framingham scores.

Previous studies reported an association of NAFLD severity with increased CIMT and carotid atherosclerotic plaques [3,35]. We found a high prevalence of increased CIMT and carotid plaques in both groups, regardless of the degree of the liver fibrosis. Large population studies such as the MESA have revealed a positive association between liver steatosis and the calcium score, independent of other cardiovascular risk factors [36,37]; however, to our knowledge, the association of calcium score and liver fibrosis has not been studied before. We found a higher median calcium score in patients with advanced liver fibrosis; however, there was no association between calcium score and liver fat content. This finding may be explained by a process called ‘burnt-out nonalcoholic steatohepatitis’, described in patients with advanced liver disease. In this process, fibrosis progression is associated with the loss of steatosis and histological inflammation [38,39]. This phenomenon explains why patients in the group with advanced fibrosis demonstrated lower fat content in spectroscopy analysis.

The mean E/e’ ratio was higher in the group with advanced fibrosis, although the atrial volume did not differ between groups in this study; these findings suggest progression to diastolic cardiac dysfunction. In a cross-sectional study involving more than 200 patients, Mantovani et al. [40] reported that NAFLD was associated with an approximately threefold increased risk of LV diastolic dysfunction, even after adjustment for traditional CVD risk factors and echocardiographic parameters. Several diastolic function indexes are worse in NAFLD; patients with the disease have lower E/A ratios and e’ velocities, higher E/e’ ratios, and larger left atrial volumes [41].

The patients in our study had neither LV nor right ventricular systolic dysfunction nor GLS reduction. Participants in the multicenter longitudinal Coronary Artery Risk Development in Young Adults study with NAFLD exhibited a subclinical alteration of systolic function (i.e., strain) but normal EFs, independent of several metabolic variables [41]. In another study, strain, assessed via MRI for the quantification of myocardial deformation in patients undergoing heart transplantation, was not associated with histological findings [42]. This lack of association between myocardial fibrosis and myocardial systolic strain may be related to variation in clinical stages, medical treatments, and, in our case, acquisition methods used for tissue tracking.

The relationship between NAFLD and incident CVD events has been observed in several studies [37], but the question of whether the prognostic value of NAFLD for CVD development is restricted to nonalcoholic steatohepatitis or is also associated with isolated steatosis remains unresolved. Liver biopsy has been used to evaluate NAFLD in some small studies, which have also revealed a significant, graded relationship between LV dysfunction and the histological severity of NAFLD, suggesting that steatohepatitis and liver fibrosis are risk factors for the development and progression of cardiovascular damage [40]. In this study, we conducted an in-depth evaluation of patients with NAFLD by examining myocardial fibrosis and found that there is an association between liver and cardiac fibrosis in these patients.

Our findings showed that myocardial ECV correlates positively with MRE liver stiffness in patients with NAFLD, with more myocardial fibrosis observed in those with more advanced liver disease. The fact that chronic inflammation is a common feature in both cardiovascular and liver disease may explain this interesting association. The quantification of the myocardial ECV is an important, consolidated, and early prognostic marker for a wide range of inflammatory and infiltrative cardiac conditions and can be used to identify heart failure with a reduced or preserved EF [43]. The presence of this marker of cardiac vulnerability is associated with poor outcomes such as heart failure and death; early identification is very important, because it can be reversible [10,44].

This study has some limitations. The sample was small, which limited the power of some analyses; in addition, our cohort was from a single tertiary center specialized in liver disease and this could limit the generalizability of our findings. Ideally, we could have included healthy controls, even though our aim was to compare two groups of patients with NAFLD with different grades of liver fibrosis. We lacked a histological validation of the ECVs in our study, but the use of the ECV, as a metric of interstitial fibrosis, has been repeatedly validated in other studies [10,45]. Future studies with larger populations are necessary to confirm our findings.

These results suggest that ECV can be used to improve risk stratification, assess disease activity, and evaluate prognosis in patients with NAFLD. Future studies are necessary to further evaluate the role of this biomarker to monitor disease progress and even guide treatment in these patients. In conclusion, patients with NAFLD may have a higher risk of future cardiovascular events, as greater degrees of liver fibrosis correlate with greater cardiac interstitial fibrosis and, hence, worse prognoses. Thus, the identification of high-risk patients with NAFLD who lack cardiac symptoms is crucial for the definition of therapeutic goals and early intervention for more effective prevention.

## Data Availability

The raw data supporting the conclusions of this article will be made available by the authors, without undue reservation.

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
