# Peer review of "How Cardiac Fibrosis Assessed via T1 Mapping Is Associated with Liver Fibrosis in Patients with Non-Alcoholic Fatty Liver Disease"

_jcm, 2023, doi:10.3390/jcm12237381_

Round 1

Reviewer 1 Report

Comments and Suggestions for Authors

This excellent article investigates the association between myocardial fibrosis and liver fibrosis in patients with nonalcoholic fatty liver disease (NAFLD). They found that patients with significant liver fibrosis had higher myocardial extracellular volume (ECV), suggesting a correlation between liver and cardiac fibrosis. I have very few comments, in no order of magnitude.

·        The absence of histological validation for ECV measurements is a limitation.

·        The absence of healthy control is a big limitation of the study.

·        If possible, authors should evaluate the presence of myocardial fibrosis with any blood biomarker, such as PICP. Having this strengthens the study's findings.

·        Authors should make a visual summary.

Reviewer 2 Report

Comments and Suggestions for Authors

The manuscript presents data on patients with NAFLD by examining myocardial fibrosis. It found an association between liver and cardiac fibrosis in T1 mapping.

Overall, it is interesting to compare hepatic MRI with cardiac MRI parameters and echocardiographic findings, and I congratulate the authors for their analyses.

However, several aspects limit the current manuscript, including an incomplete presentation of the study cohort and a small sample size.

My specific comments are:

-       The study mentions NAFLD as a referral, but the specific characteristics of the patients are not elucidated. It would be crucial to get more clinical data on these patients.

-       The study does not delve into potential variations in symptoms among the patients. Did the patients differ in their symptoms?

-       Notably, laboratory parameters, particularly NT-proBNP levels, are omitted. NT-proBNP is a well-established biomarker for heart failure and could offer valuable insights into the cardiac status of the patients. Including this data could enhance the study's diagnostic precision and contribute to a more comprehensive understanding of the relationship between NAFLD and cardiac function.

-       It's unclear whether patients with cardiac amyloidosis have been ruled out, as amyloidosis can affect both the heart and the liver.

-       Since right ventricular function is of paramount importance in conditions with preserved left ventricular ejection fraction, it is a major drawback that RVEF data are not presented. Please add data on RVEF and RVEDV.

-       The manuscript mentions increased ventricular stiffness (E/E´) associated with hepatic T1 times. To further elaborate on this connection, parameters such as vena cava backflow, and dilatation of the inferior vena cava would provide a more nuanced understanding.

-       Is data on atrial fibrillation available?

-       While fibrosis is discussed in the manuscript, investigating whether steatosis (fatty degeneration) correlates with the Framingham risk score could shed light on the interplay of the risk factors. How would you reconcile the higher median calcium value in patients with advanced liver fibrosis and not with steatotic values?

-       The sample size is very small and, therefore, not particularly reliable.

Reviewer 3 Report

Comments and Suggestions for Authors

The study describes an association between liver fibrosis and cardiac fibrosis as measured by MRI. However with a relatively small sample size, we are unable to make decisive conclusion from it. 

The main problem with this study is the patient group studied. The patients were not recruited consecutively. There was no pre defined sample size calculation. Therefore a huge bias is introduced and makes the interpretation of result extremely tricky. It is impossible to derive the association as the authors claim from this study due to this bias. 

Round 2

Reviewer 3 Report

Comments and Suggestions for Authors

The authors have tried to answer the concerns raised. Although some of the concerns remain, I am happy for the manuscript to be accepted at this stage.